# Inhibition of SARS-CoV-2 viral entry upon blocking N- and O-glycan elaboration

Qi Yang[1†], Thomas A Hughes[1†], Anju Kelkar[1†], Xinheng Yu[1], Kai Cheng[1], Sheldon Park[1], Wei-Chiao Huang[2], Jonathan F Lovell[1,2], Sriram Neelamegham[1,2,3,4]*

[1]Chemical & Biological Engineering, State University of New York, Buffalo, United States; [2]Biomedical Engineering, State University of New York, Buffalo, United States; [3]Medicine, State University of New York, Buffalo, United States; [4]Clinical & Translational Research Center, Buffalo, United States

**Abstract** The Spike protein of SARS-CoV-2, its receptor-binding domain (RBD), and its primary receptor ACE2 are extensively glycosylated. The impact of this post-translational modification on viral entry is yet unestablished. We expressed different glycoforms of the Spike-protein and ACE2 in CRISPR-Cas9 glycoengineered cells, and developed corresponding SARS-CoV-2 pseudovirus. We observed that N- and O-glycans had only minor contribution to Spike-ACE2 binding. However, these carbohydrates played a major role in regulating viral entry. Blocking N-glycan biosynthesis at the oligomannose stage using both genetic approaches and the small molecule kifunensine dramatically reduced viral entry into ACE2 expressing HEK293T cells. Blocking O-glycan elaboration also partially blocked viral entry. Mechanistic studies suggest multiple roles for glycans during viral entry. Among them, inhibition of N-glycan biosynthesis enhanced Spike-protein proteolysis. This could reduce RBD presentation on virus, lowering binding to host ACE2 and decreasing viral entry. Overall, chemical inhibitors of glycosylation may be evaluated for COVID-19.

*For correspondence:
neel@buffalo.edu

†These authors contributed equally to this work

## Introduction

SARS-CoV-2 is the virus causing the global pandemic 'coronavirus diseases 2019' (COVID-19). This is a zoonotic beta-coronavirus from bats that causes severe acute respiratory syndrome (*Lu et al., 2020*). Its effects on human physiology extends well beyond the lung as it unleashes a cytokine storm to alter immune response (*Chen et al., 2020*) and cause disseminated intravascular coagulation (*Lillicrap, 2020*). It also exhibits multiorgan tropism that impacts kidney, liver, heart and brain function (*Puelles et al., 2020*). Among its structural elements, the SARS-CoV-2 Spike protein is critical for viral attachment, fusion and entry into host (*Hoffmann et al., 2020a*; *Li et al., 2003*; *Lan et al., 2020*). The RBD region of Spike enables binding to its primary human cellular receptor, angiotensin-converting enzyme-2 (ACE2), which is ubiquitously expressed on epithelial, endothelial and blood cells (*Li et al., 2003*; *Hamming et al., 2004*). A role for heparan sulfates in viral entry has also been proposed (*Clausen et al., 2020*). Once bound, viral fusion and entry depends on two proteolysis sites, the furin/RRAR-S site located between the Spike S1 and S2 subunits, and a second S2'/SKR-S site (*Belouzard et al., 2009*). A variety of enzymes including TMPRSS2 (transmembrane protease, serine 2) and cathepsin-L may aid viral entry (*Hoffmann et al., 2020b*; *Matsuyama et al., 2005*; *Shang et al., 2020*). While inhibitors of viral binding, RNA transcription and protease activity are being tested to reduce viral load, this manuscript suggests an orthogonal approach based on glycan engineering.

Both the viral Spike-protein and ACE2 receptor are extensively glycosylated, with a majority of the 22 N-glycosylation sites of Spike and 7 N-glycosylation sites of ACE2 bearing carbohydrates (*Walls et al., 2020*; *Wrapp et al., 2020*; *Shajahan et al., 2020a*; *Figure 1*). The exact distribution of

**eLife digest** COVID-19 is an infectious disease caused by the virus SARS-CoV-2. To access the internal machinery necessary for its replication, the virus needs to latch onto and then enter host cells. Such processes rely on specific 'glycoproteins' that carry complex sugar molecules (or glycans), and can be found at the surface of both viruses and host cells. In particular, the viral 'Spike' glycoprotein can attach to human proteins called ACE2, which coat the cells that line the inside of the lungs, heart, kidney and brain. Yet the roles played by glycans in these processes remains unclear.

To investigate the role of Spike and ACE-2 glycans, Yang et al. designed a form of SARS-CoV-2 that could be handled safely in the laboratory. How these viruses infect human kidney cells that carry ACE2 was then examined, upon modifying the structures of the sugars on the viral Spike protein as well as the host ACE2 receptor. In particular, the sugar structures displayed by the virus were modified either genetically or chemically, using a small molecule that disrupts the formation of the glycans. Similar methods were also applied to modify the glycans of ACE2. Together, these experiments showed that the sugars present on the Spike protein play a minor role in helping the virus stick to human cells.However, they were critical for the virus to fuse and enter the host cells. These findings highlight the important role of Spike protein sugars in SARS-CoV-2 infection, potentially offering new paths to treat COVID-19 and other coronavirus-related illnesses. In particular, molecules designed to interfere with Spike-proteins and the viral entrance into cells could be less specific to SARS-CoV-2 compared to vaccines, allowing treatments to be efficient even if the virus changes.

the oligomannose, hybrid, complex, sialylated, and fucosylated structures in these macromolecules is likely dictated both by the protein structure and host expression system (*Shajahan et al., 2020b*; *Watanabe et al., 2020*). O-linked glycans are also reported on both proteins (*Shajahan et al., 2020a*; *Shajahan et al., 2020b*). A variety of functions have been proposed for these glycans including immunological shielding (*Watanabe et al., 2020*), direct regulation of Spike-ACE2 binding (*Bernardi et al., 2020*), and control of Spike up/down conformation (*Casalino et al., 2020*).

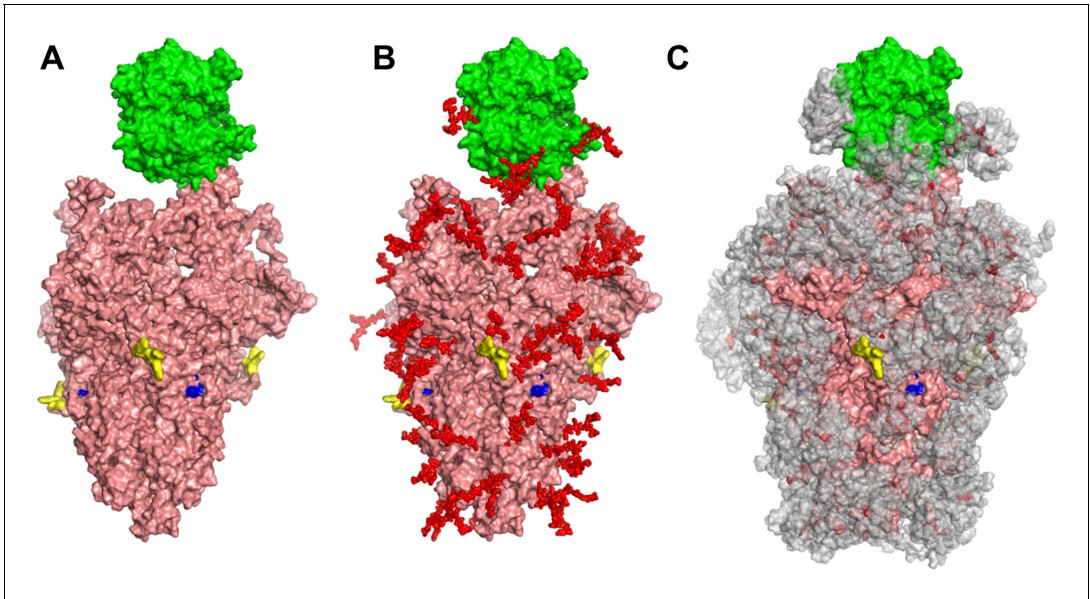

**Figure 1.** Glycan coverage of Spike-ACE2 co-complex. SARS-CoV-2 Spike protein trimer (pink) bound to ACE2 (green). (**A**) Without glycans. (**B**) With N-glycans (red) identified using LC-MS on Spike and ACE2. (**C**) Molecular dynamics simulation analyzed the range of movement of each glycan. The space sampled by glycans is represented by a gray cloud. Glycans cover the Spike-ACE2 interface. They also surround the putative proteolysis site of furin ('S1-S2', yellow) and S2' (blue).

Experimental data supporting these concepts is currently lacking and thus our understanding of the impact of glycosylation on SARS-CoV-2 function is incomplete.

We address the above gaps in knowledge in the current manuscript. Our results suggest that both the O- and N-linked glycans of the SARS-CoV-2 Spike protein, including sialylated glycan epitopes, have a relatively minor role in regulating direct Spike-ACE2 binding interactions. However, these glycans control the rates of viral entry into HEK293T cells expressing human ACE2. Here, blocking N-glycosylation on SARS-CoV-2 pseudovirus at the high-mannose stage using CRISPR-Cas9 and also a small molecule inhibitor (kifunensine) resulted in extensive cleavage/shedding of the viral Spike protein at the time of production due to enhanced proteolysis at the S1-S2 interface. Our data also suggest that glycans may contribute to additional aspects of Spike proteolysis during viral entry. Due to these effects, viral entry into human ACE2 expressing cells was reduced by >95% when virus was produced in the CRISPR-Cas9 MGAT-1 knockout cells lacking complex N-glycans, and by 85–90% upon production in the presence of kifunensine. These observations support the need to screen chemical inhibitors of glycosylation for the treatment of COVID-19.

## Results

### Modest role for ACE2 sialic acids during Spike-protein molecular recognition

In order to study the impact of N- and O-linked glycosylation on SARS-CoV-2 function, we engineered cells over-expressing full-length Spike-protein and ACE2 (*Figure 2A*). We also histidine-tag purified soluble, dimeric Fc-his fusion proteins for the Spike S1 region, RBD and extracellular portion of ACE2. The molecules were extensively glycosylated with their apparent molecular mass being greater than their theoretical mass based on peptide alone. Thus, RBD-Fc was ~60 kDa instead of a theoretical mass of 51 kDa, S1-Fc was ~150 kDa rather than 101 kDa, and ACE2-Fc was ~140 kDa instead of 110 kDa (*Figure 2B*). RBD-Fc and S1-Fc readily bound to HEK293T cells upon human ACE2 over-expression, confirming that they are functionally active (*Figure 2C*). ACE2-Fc also specifically bound Spike-protein expressed on 293Ts. Baseline RBD-Fc and S1-Fc binding to Spike expressing 293 T cells (i.e. 293 T/S) was even lower than that of wild-type 293T (red line, *Figure 2C*). This suggests expression of low amounts of Spike binding proteins on WT-293Ts. These basal Spike receptors may engage cell-surface Spike on 293 T/S cell (via *cis*-interactions) causing RBD-Fc and S1-Fc binding to be lower in 293 T/S cells compared to WT-293Ts.

Sialidase treatment of Spike-protein expressed on 293Ts did not affect ACE2-Fc binding (*Figure 2D*). Sialidase treatment of ACE2 expressed on 293Ts, however, increased RBD-Fc and S1-Fc binding by 26% and 56% respectively. Control lectin binding studies confirmed the high activity of the sialidase enzyme used in this study (*Figure 2—figure supplement 1A*). The relatively small effect of the sialidase in these studies was not due to the absence of sialic acid on either S1-Fc or ACE2-Fc. In this regard, independent glycoprotemics mass spectrometry (MS) data showed that ~80–100% of the N-glycans expressed at specific sites of ACE2-Fc and ~40–100% of the S1-Fc glycopeptides expressed complex-type glycans (K.C., et al., manuscript in preparation). Up to ~60% of the antennae on these complex structures on ACE2 and ~20% of the structures on S1-Fc were terminated by sialic acid.

### Neither sialidase treatment of Spike nor ACE2 markedly impacted viral entry

To complement the above binding studies, we measured SARS-CoV-2 pseudovirus entry into stable 293T/ACE2 cells. This assay provides an aggregate measure of both molecular binding and viral fusion/entry in the context of the physiological Spike trimeric configuration (*Tai et al., 2020*). Here, the control VSVG (*Vesicular stomatitis* virus G-protein) pseudotyped virus displayed broad tropism both for wild-type HEK293T and stable 293T/ACE2 (*Figure 2E*, *Figure 2—figure supplement 1B*). Wild-type SARS-CoV-2 Spike-protein virus ('Spike-WT') only entered 293T/ACE2 cells confirming the strict dependence on cell-surface ACE2 receptor. A 'Spike-mutant' pseudovirus, where the 'RRAR' furin-site was swapped with an 'SRAS' sequence, also only entered 293T/ACE2 (22). This mutation results in reduced proteolysis at the S1-S2 interface, as discussed later, likely due to the presence of a single arginine site at the S1-S2 interface rather than the polybasic ('RRAR') sequence. To study

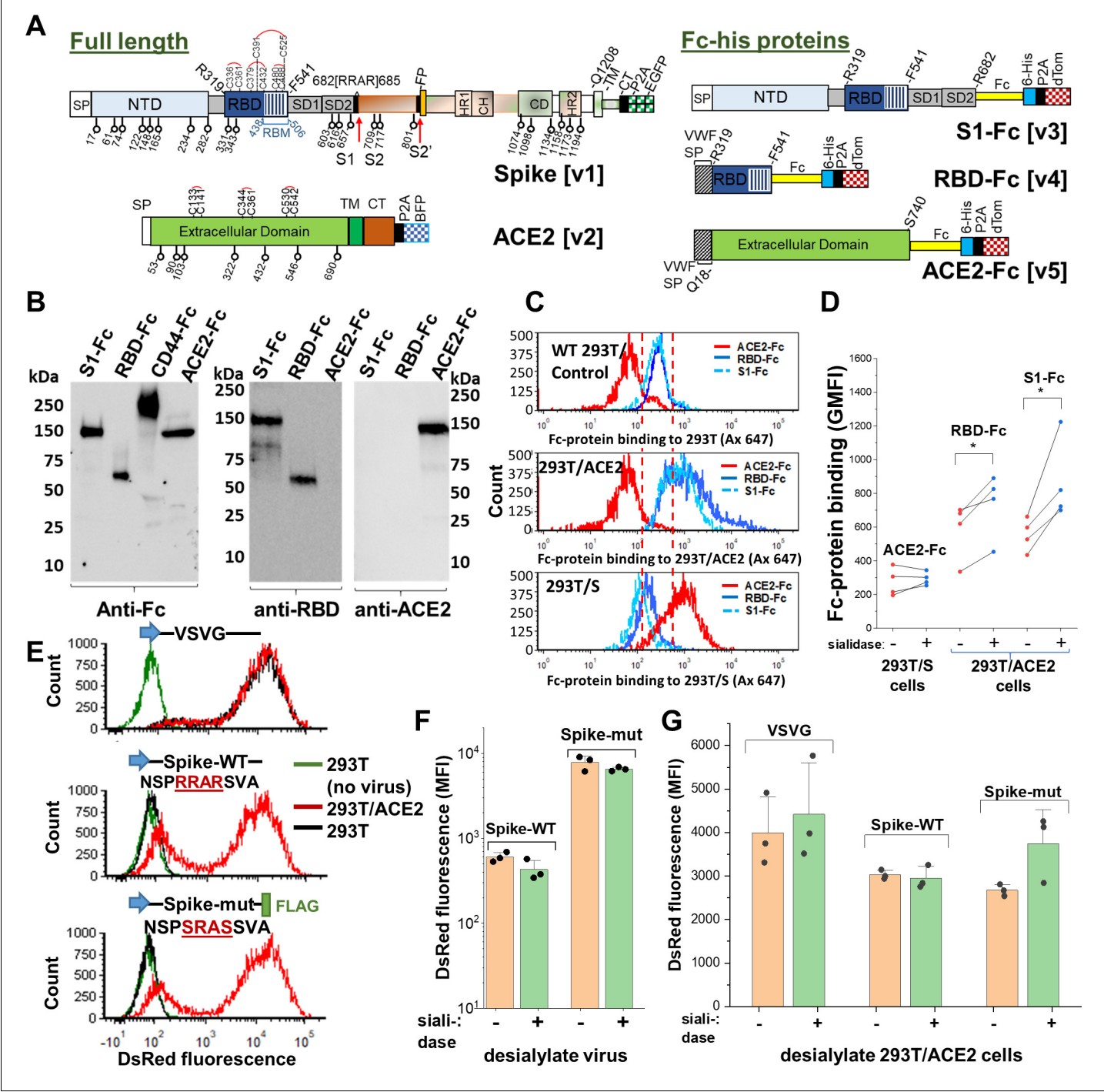

**Figure 2.** Sialic acid has modest effect on Spike binding and viral entry. (**A**) Full-length proteins expressed on cells include wild-type Spike-protein [v1] and human ACE2 [v2]. N-glycosylation sites are indicated by lollipop. Fc-his soluble proteins encode for S1-subunit [v3], RBD [v4] and soluble ACE2 [v5]. All constructs were co-expressed with fluorescent reporters separated by P2A. Note that the Fc-section also contains one N-glycosylation site. (**B**) Western blot for purified Fc-proteins from HEK293T probed with anti-Fc, anti-RBD or anti-ACE2 Ab. CD44-Fc is positive control. (**C**) Flow cytometry data showing S1-Fc (1.7 µg/mL) and RBD-Fc (0.35 µg/mL) binding to ACE2 expressed on HEK293T (middle panel). Spike expression enhances ACE2-Fc (1.4 µg/mL) binding (bottom). (**D**) Desialylation of Spike-protein expressed on 293 T/S had minimal effect on ACE2-Fc (0.7 µg/mL) binding. ACE2 desialylation on 293T/ACE2 increased binding of RBD-Fc (0.2 µg/mL) and S1-Fc (1.7 µg/mL) by 26–56% (paired experiments, *p<0.05). (**E**) Pseudovirus with DsRed-reporter were developed with three different envelope proteins. VSVG pseudotyped virus infected both HEK293T (black line) and stable 293T/ACE2 (red line) cells. Virus with Spike-WT and Spike-mutant entered 293T/ACE2 only. (**F**) Same titer of virus (0.3 µg/mL p24-equivalent) were treated with or without sialidase, prior to addition to stable 293T/ACE2 cells. Infection using Spike-mutant was higher compared to Spike-WT. Sialidase

*Figure 2 continued on next page*

*Figure 2 continued*

treatment of virus had no effect. (G) 293T/ACE2 cells were sialidase treated prior to addition of VSVG (0.3 µg/mL p24-equiv.), Spike-WT (1.5 µg/mL p24-equiv.) or Spike-mutant (0.2 µg/mL p24-equiv.) pseudovirus. Sialidase treatment did not affect viral entry. Abbreviations: Spike signal peptide (SP), N-terminal domain (NTD), receptor-binding domain (RBD), receptor-binding motif (RBM), subdomain 1 (SD1), subdomain 2 (SD2), fusion peptide (FP), heptad repeat 1 (HR1), central helix (CH), connector domain (CD), heptad repeat 2 (HR2) transmembrane section (TM), cytoplasmic tail (CT), ACE2: Angiotensin-converting enzyme-2; VSVG: *Vesicular stomatitis* virus G-protein; WT: wild-type; mut: mutant.

The online version of this article includes the following figure supplement(s) for figure 2:

**Figure supplement 1.** Sialidase treatment studies.
**Figure supplement 2.** p24 assay to determine viral titer.

the role of sialic acid on viral entry, we titered the Spike-WT and Spike-mutant virus using p24 ELISA (*Figure 2—figure supplement 2*). The viruses were then treated with a pan-sialidase from *Arthrobacter ureafaciens*, and the same amount of Spike-WT and Spike-mutant particles were applied to stable 293T/ACE2 cells (*Figure 2F*, *Figure 2—figure supplement 1C*). Here, Spike-mutant pseudotyped virus was ~5–10 times more effective at stimulating DsRed-reporter expression compared to Spike-WT. Thus, the efficiency of cleavage at the S1-S2 interface can fine-tune SARS-CoV-2 infectivity (*Hoffmann et al., 2020b*; *Shang et al., 2020*; *Walls et al., 2020*). Sialidase treatment of virus did not impact viral entry, consistent with the notion that Spike sialic acid does not regulate molecular recognition. Additionally, sialidase treatment of 293T/ACE2 did not alter either VSVG or Spike-WT pseudovirus entry (*Figure 2G*, *Figure 2—figure supplement 1D*). A partial increase in Spike-mutant viral entry was measured upon sialidase treatment (p=0.08), suggesting a minor inhibitory role for ACE2 sialic acid when viral infectivity is high. Glycoproteomics analysis of purified pseudovirus produced in HEK293T cells showed that ~45–100% of the viral N-glycans were complex-type, with up to ~55% of their antennae being terminated by sialic acid (K.C., et al. manuscript in preparation). Thus, the pseudovirus used in this study was processed by glycoenzymes that are typically found in the cellular medial and trans-Golgi compartments. Together, the binding and pseudovirus data suggest that ACE2 sialic acids may modestly shield virus/Spike binding in some biological contexts and perhaps some cell systems.

## Blocking Spike-protein N-glycan biosynthesis at the high-mannose stage partially reduced ACE2 binding

To determine if other aspects of N- and O-linked glycosylation may impact SARS-CoV-2 function, we applied CRISPR-Cas9 technology to develop isogenic HEK293T clones that lack either the core-1 O-glycan forming galactosyltransferase C1GalT1 or the N-glycan branching β1,2GlcNAc-transferase MGAT1 (*Figure 3A*; *Stolfa et al., 2016*; *Chugh et al., 2018*). Here, knocking out C1GalT1 blocks O-glycan biosynthesis at the GalNAcα-Ser/Thr (+/- sialic acid) stage as it is not elaborated to form the core-1 structure (Galβ1,3GalNAcα). These cells are termed '[O]⁻293T'. Knocking out MGAT1 blocks N-glycan biosynthesis at the Man-5 stage, and the resulting clone is called '[N]⁻293T'. Sanger sequencing confirmed that all three C1GalT1 alleles of [O]⁻293T contained indels (*Figure 3B*). Frameshift mutation was also noted at the single MGAT1 allele in [N]⁻293T. As anticipated, VVA-lectin (*Vicia villosa agglutinin*) binding to GalNAcα was dramatically augmented in the [O]⁻293Ts (*Figure 3C*). Complex glycans were absent in [N]⁻293Ts as they did not engage PHA-L (lectin from *Phaseolus vulgaris*). Additional data with a panel of fluorescent lectins further confirm specific blockade of N-glycan complex structures and lactosamine chains in [N]⁻293T, and inhibition of O-glycan extension in the [O]⁻293 T cells (*Figure 3—figure supplement 1*). Together, the findings confirm the absence of O- and N-glycan elaboration in the [O]⁻293T/C1GalT1-KO and [N]⁻293T/MGAT1-KO cells, respectively.

Wild-type 293Ts and the glycoenzyme knockouts were transiently transfected to express Spike and ACE2 for binding studies. Both full-length proteins expressed well and at equal levels in the different cell systems (*Figure 3—figure supplement 2A*). In order to quantify the effect of glycosylation on ACE2-Spike binding, ACE2-Fc was applied to Spike bearing cells, either 293T or the O/N-glycan knockouts (*Figure 3D*). Similarly, either S1-Fc (*Figure 3E*) or RBD-Fc (*Figure 3F*) were applied to ACE2 bearing 293T or the N-/O-glycan knockouts. These Fc-proteins were applied at sub-saturation concentrations in the cytometry binding studies in order to quantitatively evaluate differences in

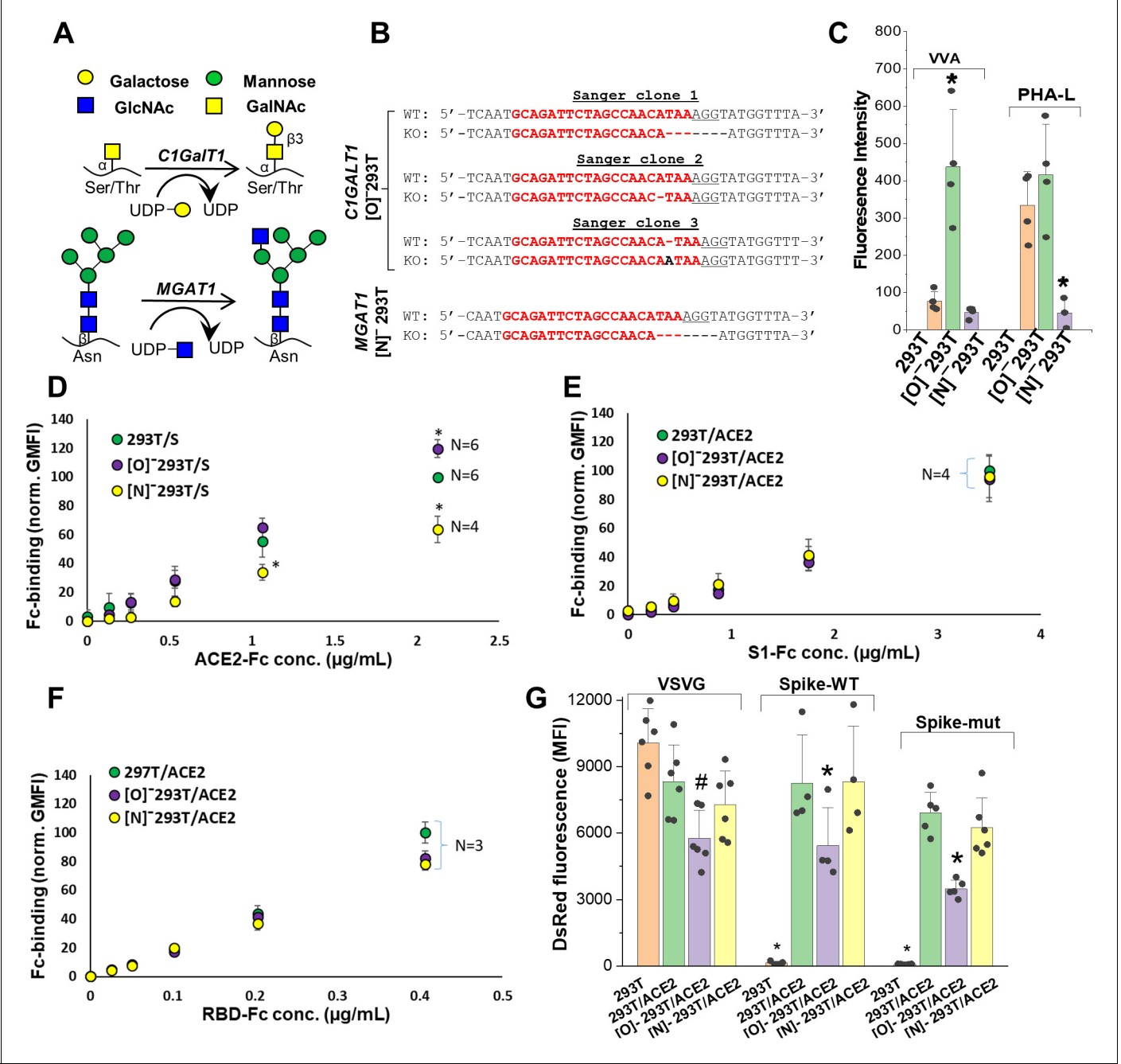

**Figure 3.** ACE2 glycosylation does not affect viral entry. (**A**) Knocking out *C1GALT1* and *MGAT1* using CRISPR-Cas9 inhibits O- and N-glycan biosynthesis in HEK293Ts. (**B**) Sanger sequencing results of isogenic 293T clones shows indels on all 3 alleles of *C1GALT1* ('[O]⁻293T') and single allele of *MGAT1* ('[N]⁻293T') knockout cells. Wild-type (WT) sequence is on the first line. Lower line shows base deletions (hyphen) and insertions (black fonts) for individual KOs. sgRNA target sequence is in red and protospacer adjacent motif is underlined. (**C**) Increased VVA and reduced PHA-L binding confirm loss of O-linked glycans in [O]⁻293Ts and N-glycans in [N]⁻293Ts, respectively. (**D**) Knocking out N-glycans on Spike protein reduced ACE2-Fc binding in cytometry based binding studies. Knocking out Spike O-glycans increased ACE-2 binding. (**E–F**) Truncation of ACE2 N- and O-glycans did not affect either S1-Fc (panel **E**) or RBD-Fc (panel **F**) binding. (**G**) ACE2 was transiently expressed on 293T, [O]⁻293T and [N]⁻293 T cells. All pseudotyped virus efficiently entered ACE2 expressing cells. Virus was not titered for these runs, and thus comparison between viruses is not possible. *p<0.05 with respect to all other treatments. #p<0.05 with respect to 293T and [N]⁻293T/ACE2 in panel **G**.

The online version of this article includes the following figure supplement(s) for figure 3:

**Figure supplement 1.** Lectin binding to wild-type and glycogene-KO 293 T cells.
**Figure supplement 2.** Effect of ACE2 glycosylation on viral entry.

receptor-ligand-binding. Here, we observed a 40% decrease in ACE2-Fc binding to cell-surface Spike expressed on [N]⁻293Ts, and a slight 20% increase when Spike was expressed on the [O]⁻293Ts (*Figure 3D*). Glycan engineering of ACE2 in the [O]⁻ and [N]⁻293Ts did not impact either S1-Fc (*Figure 3E*) or RBD-Fc binding (*Figure 3F*). Here, RBD-Fc bound ~10 times more avidly to ACE2 compared to S1-Fc. Together, the data suggest a modest role for Spike-protein glycosylation on direct Spike-ACE2 binding. Our observation that the binding of RBD for ACE2 was substantially higher than S1-ACE2 interactions is consistent with a previous report (*Shang et al., 2020*). This highlights the need to carefully consider RBD presentation/conformation in the context of the full protein when quantifying molecular affinity to ACE2.

## Blocking N-glycan elaboration on Spike abrogated viral entry

We evaluated the impact of ACE2 and Spike glycosylation on viral entry. To determine the impact of ACE2 glycosylation, we transiently expressed this protein on wild-type 293Ts, [O]⁻293Ts and [N]⁻293Ts. The entry of the pseudovirus (VSVG, Spike-WT and Spike-mutant) to these three cell types was then evaluated (*Figure 3G*, *Figure 3—figure supplement 2B*). Here, the pattern of viral entry was similar in all cell systems. Lower entry was measured for Spike-WT and Spike-mutant pseudovirus infection of [O]⁻293T/ACE2 cells, but this was also noted for the control VSVG-virus. Thus, this reduced infectivity in the O-glycan knockout is not exclusively due to ACE2 glycans. We conclude that ACE2 glycans may not regulate viral entry.

In contrast to ACE2, glycan engineering of viral Spike protein dramatically affected viral entry (*Figure 4*). To study this, VSVG, Spike-WT and Spike-mutant pseudotyped virus were generated in wild-type 293T, [O]⁻293T and [N]⁻293T (*Figure 4A*). All nine viruses were used to infect stable 293T/ACE2 cells. When produced in [O]⁻293T, Spike-WT and Spike-mutant pseudovirus resulted in a 70–85% loss in DsRed fluorescence (*Figure 4B*), and a partial reduction (35–70%) in the fraction of DsRed positive cells (*Figure 4C*). In stark contrast, knocking out N-glycans in both pseudoviruses abrogated viral infection into 293T/ACE2 cells by >95% (*Figure 4B and C*). Dose studies confirmed loss of viral function upon inhibiting N-glycan elaboration over a range of viral titers (*Figure 4D*). This was observed for the Spike-mutant and also the Spike-WT pseudovirus (data not shown). Western blot with anti-S2 pAb (polyclonal antibody) suggest that both the Spike-WT and Spike-mutant virus expressed comparable levels of Spike protein, regardless of the glycosylation mutation (*Figure 4E*). The molecular mass of Spike expressed in [N]⁻293T was lower than that of other virus consistent with the extensive N-glycosylation of this protein. Additionally, the data suggest more extensive cleavage of Spike-WT compared to Spike-mutant, consistent with the presence of the proteolysis-sensitive polybasic furin-site in Spike-WT. The Spike pseudovirus produced in [N]⁻293Ts also appeared to have lower intact Spike protein. To examine this more closely as the anti-S2 pAb may have preference for binding the S2-domain over full Spike, we measured the FLAG-epitope at the C-terminus of Spike-mutant using an anti-FLAG mAb (monoclonal antibody). This may provide more uniform recognition of both the full Spike and the S2-subunit (*Figure 4F*). This blot suggests that N-glycan loss may trigger precipitous (95%, based on densitometry) loss of intact Spike protein due to enhanced cleavage at the S1-S2 interface. Consistent with the viral entry assay, we also noted greater proteolysis of virus produced in [O]⁻293Ts (52%) compared to that produced in wild-type 293Ts (32%). Together, the data suggest that both Spike N-glycans, and possibly also O-glycans, may play a role in regulating SARS-CoV-2 Spike-protein stability.

## Small molecule inhibitors of N-linked glycosylation drastically reduced viral entry

Remarkably, the same observations as seen in the viral N-glycan knockouts could be reproduced upon producing viral particles in cells cultured with the potent mannosidase-I alkaloid inhibitor kifunensine ($K_I$ ~25–125 nM, *Elbein et al., 1990*; *Figure 5A*). This water-soluble inhibitor reduced the formation of complex N-glycans on cultured cells within 24–48 hr with no change in cell viability (*Figure 5—figure supplement 1A*). Virus produced in culture with 15 µM kifunensine displayed ~85–90% reduction in infection, as measured using DsRed-reporter in microscopy (*Figure 5B*, *Figure 5—figure supplement 1B*) and cytometry assays (*Figure 5C and D*). The reduction in Spike molecular mass upon addition of this small molecule was less apparent compared to that in [N]⁻293Ts (*Figure 5E*), since blockade of Mannosidase-I preferentially gives rise to Man7-9 glycans while the

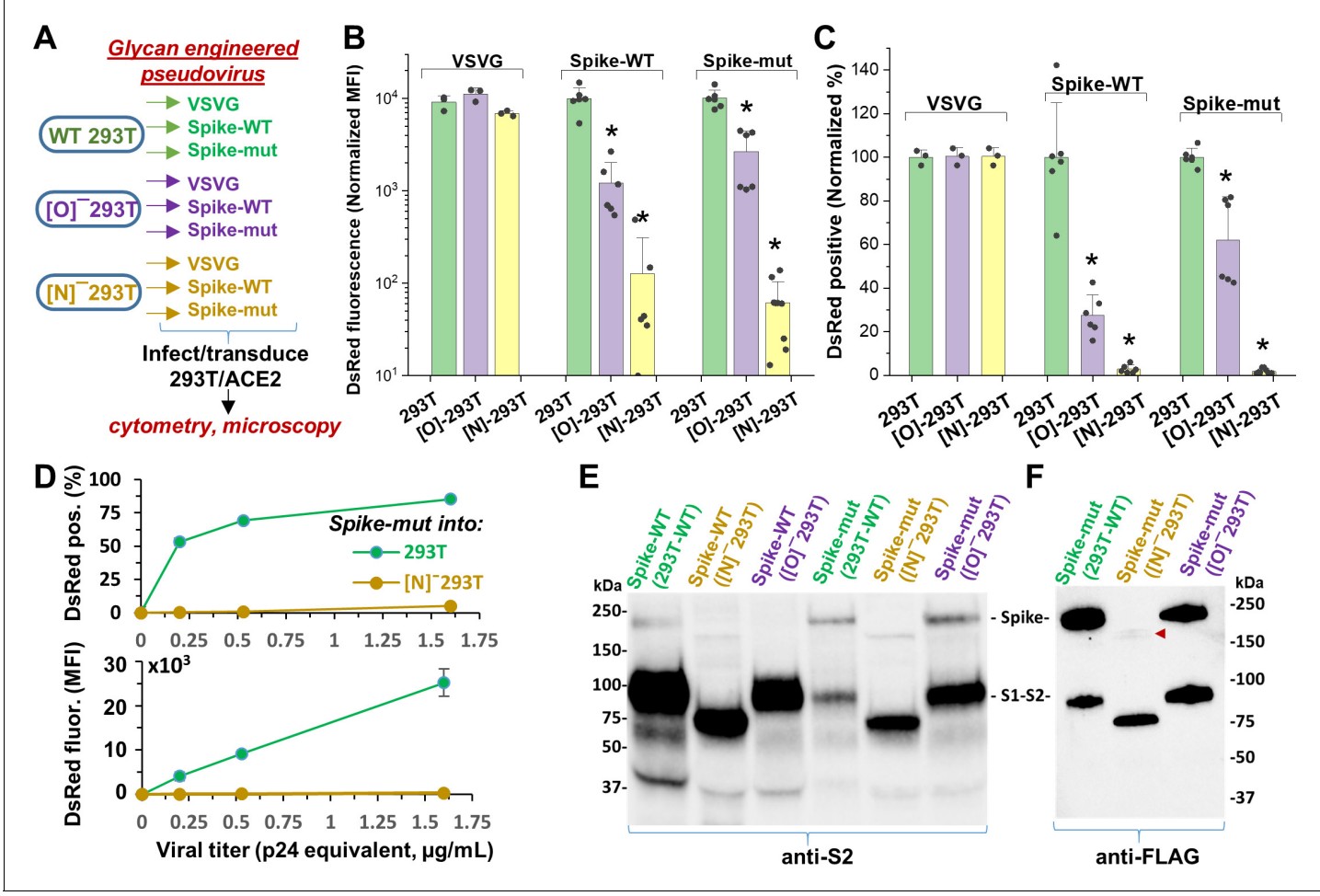

**Figure 4.** N-glycan modification of SARS-CoV-2 pseudovirus abolishes entry into 293T/ACE2 cells. (**A**) Pseudovirus expressing VSVG envelope protein, Spike-WT and Spike-mutant were produced in wild-type, [O]⁻ and [N]⁻293 T cells. All nine viruses were applied at equal titer to stable 293T/ACE2. (**B–C**) O-glycan truncation of Spike partially reduced viral entry. N-glycan truncation abolished viral entry. In order to combine data from multiple viral preparations and independent runs in a single plot, all data were normalized by setting DsRed signal produced by virus generated in wild-type 293T to 10,000 normalized MFI or 100% normalized DsRed positive value. (**D**) Viral titration study performed with Spike-mutant virus shows complete loss of viral infection over a wide range. (**E**) Western blot of Spike protein using anti-S2 Ab shows reduced proteolysis of Spike-mut compared to Spike-WT. The full Spike protein and free S2-subunit resulting from S1-S2 cleavage is indicated. Molecular mass is reduced in [N]⁻293T products due to truncation of glycan biosynthesis. (**F**) Anti-FLAG Ab binds the C-terminus of Spike-mutant. Spike produced in [N]⁻293Ts is almost fully proteolyzed during viral production (red arrowhead). *p<0.05 with respect to all other treatments.

MGAT1/[N]⁻293Ts knockouts produce Man-5. Here also, the Spike-protein proteolysis was more extensive upon kifunensine addition though it was not complete, as observed in the anti-FLAG blot.

In order to determine if the enhanced S1-S2 site proteolysis of Spike protein upon N-glycan inhibition is the exclusive reason for reduced viral entry, we produced virus with a Spike variant ('Spike-delta') that contained the C-terminal FLAG-epitope but that lacked the furin-site (*Figure 5F*). The substitution of the 'RRAR' sequence with a single Alanine ('A') resulted in viral Spike that was not extensively cleaved both in the absence and presence of kifunensine. Robust viral entry into 293T/ACE2 cells was observed using this furin resistant virus, and this too could be partially blocked by kifunensine. Thus, in addition to proteolysis at the S1-S2 site, N-glycans may have additional roles during viral entry.

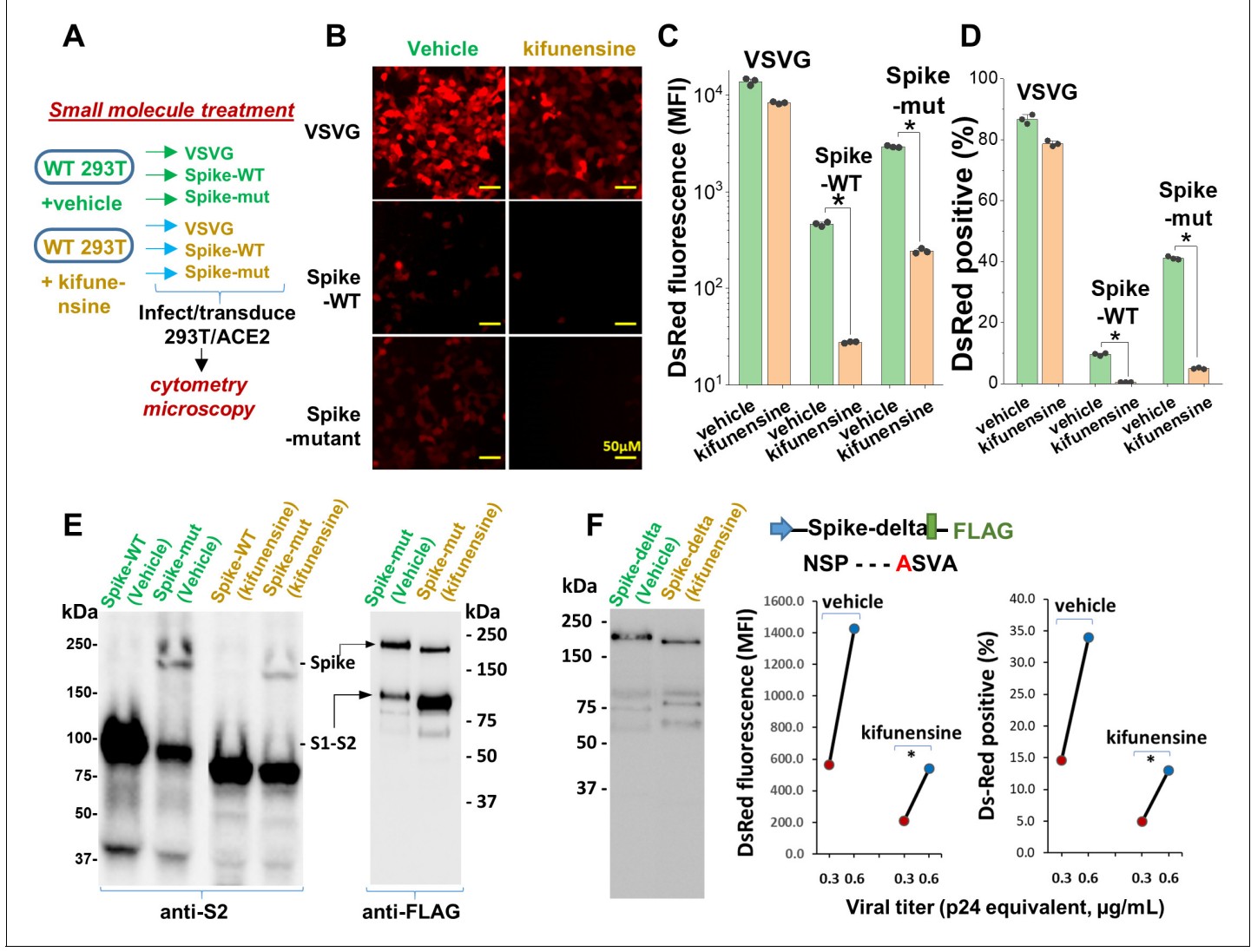

**Figure 5.** Mannosidase-I blockade using kifunensine inhibits SARS-CoV-2 pseudovirus entry into 293T/ACE2 cells. (A) VSVG, Spike-WT and Spike-mutant pseudovirus were produced in the presence of 15 µM kifunensine or vehicle control. The six viruses were added to 293T/ACE2 at equal titer. (B–D) Microscopy (panel B) and cytometry (panel C, D) show ~90% loss of viral infection in the case of Spike-WT and Spike-mutant virus upon kifunensine treatment (*p<0.05). (E) Spike molecular mass is reduced in the western blots due to high-mannose glycan synthesis in runs with kifunensine. Intact Spike is reduced in the presence of kifunensine, in anti-FLAG blot. (F) The polybasic furin 'RRAR' site was substituted by a single 'A' amino acid in Spike-delta. Virus with Spike-delta were expressed both in the presence of vehicle and kifunensine. Western blot shows lack of S1-S2 cleavage in this construct. In viral entry assay, kifunensine reduced DsRed expression in 293T/ACE2 cells, even in the case of Spike-delta pseudovirus (*p<0.05). Similar observation was made at two different viral titers (0.3 and 0.6 µg/mL p24 equivalent).

The online version of this article includes the following figure supplement(s) for figure 5:

**Figure supplement 1.** Effect of kifunensine.

## Discussion

This manuscript undertook a series of studies in order to comprehensively describe the role of O- and N-linked glycans on SARS-CoV-2 Spike binding to human ACE2 (*Figure 6A*), and related viral entry (*Figure 6B*). It demonstrates only a minor role for carbohydrates in regulating Spike-ACE2 direct binding. In this regard, we observed some enhancement in Spike binding and viral entry upon treating 293T/ACE2 with a pan-sialidase. Based on the published crystal structure and molecular modeling, the critical glycans regulating this process are likely located at Asn-90 or Asn-322, proximal to the Spike binding interface (*Lan et al., 2020*; *Figure 1*). The effect was nevertheless small compared to that reported for other sialic acid dependent viruses like influenza (*Stencel-*

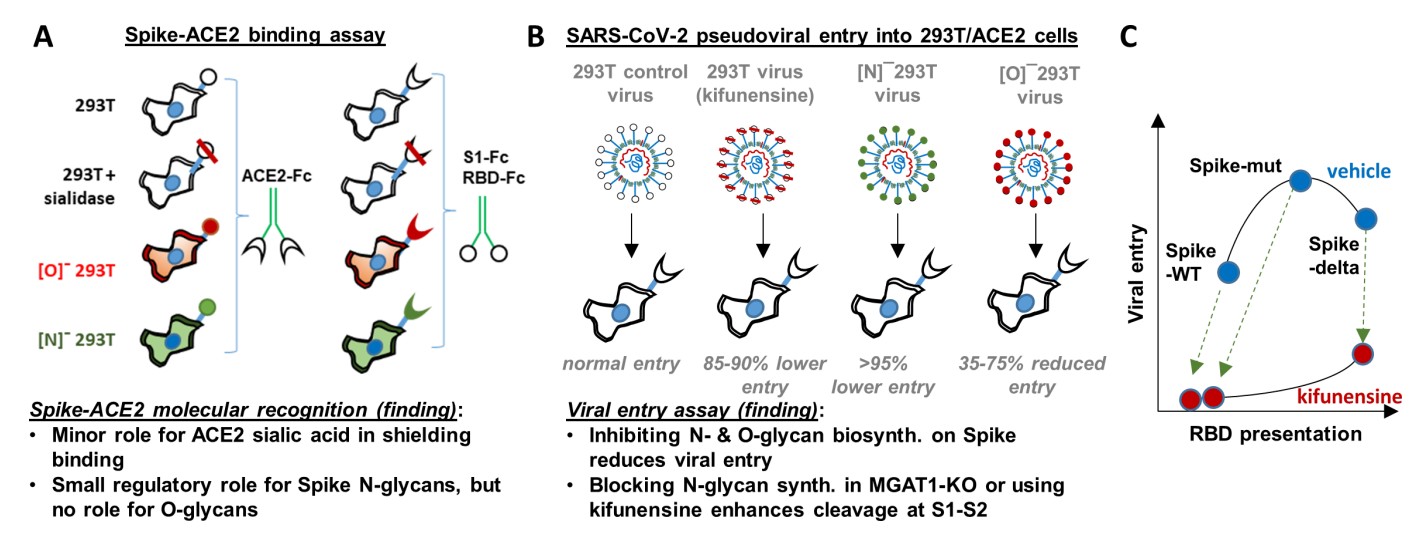

**Figure 6.** Principal findings and conceptual model. (A) ACE2-Fc binding was measured to wild-type or glycoEnzyme-KO 293 T cells expressing Spike. Sialidase treatment of cells was performed in some cases. Similar studies also measured S1-Fc and RBD-Fc binding to cell-surface expressed ACE2. (B) SARS-CoV-2 pseudovirus (bearing Spike-WT, Spike-mut, Spike-delta variants) were generated in wild-type or glycoEnzyme-KO 293Ts, in the presence and absence of kifunensine. Main results of binding (A) and viral entry (B) assay are listed. (C) Conceptual model shows that kifunensine can induce S1-S2 site proteolysis on Spike-WT and Spike-mut virus, but not Spike-delta virus. This proteolysis reduces RBD presentation and attenuates viral entry into 293T/ACE2. Without affecting S1-S2 cleavage, kifunensine also partially reduced Spike-delta pseudovirus entry function. The data suggest additional roles for Spike N-glycans during viral entry.

*Baerenwald et al., 2014*), reovirus, parovirus (*Löfling et al., 2013*) and coronavirus like the Middle-East Respiratory Syndrome virus/MERS (*Li et al., 2017*).

The studies performed with CRISPR-Cas9 cell lines containing disrupted O- and N-linked glycans also support the concept that glycosylation is not a critical regulator of ACE2-Spike binding. Here, blocking O- and N-glycan biosynthesis on ACE2 did not impact either viral entry or Spike-Fc binding. Thus, the functional change observed upon enzymatically removing sialic acid on ACE2 is likely a steric effect, as it can be offset by other modifications. Blocking elaboration of O- and N-glycans on Spike, also, only had a partial effect on ACE2-Fc binding. This observation is consistent with previous cryo-EM (*Walls et al., 2020*; *Wrapp et al., 2020*) and molecular dynamics simulation results (*Grant et al., 2020*), which show that the active/'up' form of Spike RBD is largely exposed with few glycans at the binding interface. It is now well appreciated that this lack of glycan shielding, makes the exposed RBD a prime target for vaccine development.

In contrast to the modest effect of glycosylation on receptor-ligand-binding, we observed a much more profound role for glycans in regulating viral entry in part due to their impact on Spike proteolysis at the S1-S2 interface. In this regard, we observed that blocking N-glycan elaboration using both genetic approaches (i.e. MGAT1-KO/[N]⁻293T) and the natural product mannosidase-I inhibitor, kifunensine, dramatically reduced viral entry into 293T/ACE2 cells. This was observed for two virus types: Spike-WT and Spike-mut. In the case of Spike-mut, enzymatic and small molecule inhibition resulted in enhanced proteolysis of Spike-protein during pseudovirus production. These data suggest an inverse relation between furin cleavage and viral entry into 293T/ACE2 cells. Consistent with this, Spike-mut displayed reduced S1-S2 proteolysis compared to Spike-WT, and higher viral infectivity. Others have also noted a reduction in viral entry, upon increasing furin cleavage potential by addition of more basic residues ('RRRKR' mutation) at the S1-S2 interface (*Hoffmann et al., 2020b*). Finally, a recent study observed that the introduction of the D614G mutation in Spike reduced furin cleavage, and this resulted in enhanced viral infectivity (*Korber et al., 2020*). SARS-CoV-2 containing the D614G mutation is currently the dominant strain worldwide (>80% prevalence, *Zhang et al., 2020*), and the possibility that this is due to altered furin cleavage potential is hotly debated. Overall, our studies with MGAT1-KO, C1GalT1-KO and kifunensine suggest that both N- and O-glycans

may modulate the rates of proteolysis at the S1-S2 interface, thus affecting viral entry (*Figure 6C*). While some S1 may be associated with Spike even after partial cleavage at the S1-S2 interface (*Belouzard et al., 2009*), this may potentiate a quantitative reduction in RBD presentation on viral surface, resulting in reduced ACE2 binding and lower viral entry. Indeed several N-glycans are located proximal to the furin-site: $N^{603}$ and $N^{657}$, and also $N^{61}$ (*Watanabe et al., 2020*). O-glycans have also been hypothesized to exist in this region at $S^{673}$, $T^{678}$ and $S^{686}$ (*Andersen et al., 2020*), and some of these have been experimentally verified (*Sanda et al., 2020*). How this site-specific glycosylation affects SARS-CoV-2 entry, remains to be determined.

In addition to its role in regulating RBD presentation, our data suggest that N-glycans may also have other roles in regulating viral entry. In this regard, we noted that even the robust entry of Spike-delta pseudovirus into 293T/ACE2 cells could be partially inhibited by kifunensine (*Figure 6C*), and this is independent of S1-S2 proteolysis. Others have noted that the effect of 'Spike-delta' mutation may be cell type specific in that this mutation results in lower infectivity in some cell types like Calu-3 which rely on the enzyme TMPRSS2 for viral fusion, while this mutation does not affect entry into other cell types like VeroE6 which lacks TMPRSS2 (*Hoffmann et al., 2020b*). Based on these observations, we speculate that additional glycosylation sites proximal to S2' may also modulate viral entry, perhaps because some proteases like TMPRSS2 have both ligand-binding and protease activity. The N-glycans proximal to S2' include $N^{801}$, and to a lesser extent $N^{616}$ and $N^{657}$. Additional studies with more cell types and mutant virus are necessary in order to fully elucidate the molecular mechanism by which O- and N-glycans regulate host-cell specific viral entry response.

In addition to basic science understanding, our findings have translational potential as it suggests that both O- and N-glycan truncation may be used to reduce viral entry. In this regard, it would be attractive to test additional N-glycosylation inhibitors, like Swainsonine, which has a demonstrated safety profile in humans (*Goss et al., 1997*). Additional potential inhibitors that may modulate viral entry include α-mannosidase inhibitors (deoxymannojirimycin, mannostatin A), α-glucosidase inhibitors (N-butyl deoxynojirimycin, N-nonyl-deoxynojirimycin, castanospermine, celgosivir) (*Clarke et al., 2020*), and finally compounds like SNAP (*Del Solar et al., 2020*; *Wang et al., 2018*) and 4F-GalNAc (*Marathe et al., 2010*) which block various aspects of N- and O-glycan biosynthesis. These potentially represent off-the-shelf drugs and compounds that may be repurposed to reduce viral load and ameliorate SARS-CoV-2 related respiratory symptoms. Studies are currently underway in our laboratory to test these concepts, and extend the assay to authentic SARS-CoV-2 strains. Overall, while SARS-CoV-2 has developed a natural ability to enhance infectivity, we propose that glycoengineering may provide a strategy to take advantage of these evolutionary features to regulate viral entry and reduce disease transmission.

# Materials and methods

**Key resources table**

| Reagent type (species) or resource | Designation | Source or reference | Identifiers | Additional information |
|---|---|---|---|---|
| Antibody | Anti-RBD Spike (rabbit polyclonal) | Sino Biologicals (Beijing, China) | Cat#: 40592-T62 | WB (1:5000) |
| Antibody | Anti-S2 Spike (rabbit polyclonal) | Sino Biologicals (Beijing, China) | Cat#: 40590-T62 | WB (1:5000) |
| Antibody | Anti-human ACE2 AF933 (goat polyclonal) | R and D Systems | Cat#: AF933 RRID:AB_355722 | Binding assay (1 ul) |
| Antibody | Anti-human ACE2 (mouse monoclonal) | R and D Systems | Cat#: MAB9332 | Binding assay (1 ul) |
| Antibody | Anti-FLAG clone L5, (rat monoclonal) | Biolegend (San Diego, CA) | Cat# 637303 RRID:AB_1134265 | WB (1:5000) |

*Continued on next page*

Continued

| Reagent type (species) or resource | Designation | Source or reference | Identifiers | Additional information |
|---|---|---|---|---|
| Antibody | Anti-human F(ab')two conjugated with Alexa Fluor FITC (goat polyclonal) | Jackson ImmunoResearch | Cat# 109-096-098 RRID:AB_2337668 | Binding assay (1:150) |
| Antibody | Anti-human F(ab')two conjugated with Alexa Fluor 647 (goat polyclonal) | Jackson ImmunoResearch | Cat# 109-606-098 RRID:AB_2337899 | Binding assay (1:150) |
| Other (plant lectin) | PHA-L (Phaseolus vulgaris Leucoagglutinin, binds Galβ1, 4GlcNAcβ1, 6(GlcNAcβ1, 2Manα1,3)Manα1, 3 on complex N-glycans) | Vector Laboratories | Cat# FL-1111 RRID:AB_2336655 | Lectin binding study (1:200) |
| Other (plant lectin) | VVA-lectin (binds GalNAcα) | Vector Laboratories | Cat# AL-1233 RRID:AB_2336854 | Lectin binding study (1:200) |
| Other (plant lectin) | SNA (Sambucus nigra lectin, binds primarily α2,6 sialic acid) | Vector Laboratories | Cat# FL-1301 RRID:AB_2336719 | Lectin binding study (1:200) |
| Other (plant lectin) | ECL (Erythrina cristagalli lectin, binds desialylated lactosamine Galβ1, 4GlcNAc chains) | Vector Laboratories | Cat# L-1140 RRID:AB_2336438 | Lectin binding study (1:500) |
| Strain, strain background (HIV-1 lentivirus) | Lentivirus with DsRed reporter and various envelope proteins | This paper | | Various pseudovirus types are made in this paper by authors |
| Peptide, recombinant protein | Sialidase (Arthrobacter ureafaciens α2–3,6,8,9-Neuraminidase) | New England BioLabs | Cat#: P0722S | |
| Chemical compound, drug | Kifunensine | Toronto Research Chemicals | Cat#: K450000 | |
| Cell line (Homo-Sapiens) | HEK 293T | ATCC | Cat# KCB 200744YJ RRID:CVCL_0063 | |
| Cell line (Homo-sapiens) | HEK 293T Lenti-X | Clontech/ Takara Bio | Cat#: 632180 | |
| Cell line (Homo-sapiens) | HEK/ACE2 | Michael Farzan (Scripps Research, Jupiter, FL) | | |
| Cell line (Homo-sapiens) | [O]⁻293T | This paper | | C1GalT1 Knock out HEK 293 T cell |
| Cell line (Homo-sapiens) | [N]⁻293T | This paper | | MGAT1 Knock out HEK 293 T cell |

*Continued*

| Reagent type (species) or resource | Designation | Source or reference | Identifiers | Additional information |
|---|---|---|---|---|
| Recombinant DNA reagent | Spike-WT [v1] | Dr. Haisheng Yu (Institute of Laboratory Animal Science, Peking Union Medical College) | | Plasmid to express full length Spike |
| Recombinant DNA reagent | ACE2 [v2] | This paper | Derived from RRID:Addgene_1786 | Plasmid to express full length human ACE2 |
| Recombinant DNA reagent | S1-Fc [v3] | This paper | | Plasmid to express S1 as an human Fc IgG1 fusion protein |
| Recombinant DNA reagent | RBD-Fc [v4] | This paper | | Plasmid to express RBD as an human Fc IgG1 fusion protein |
| Recombinant DNA reagent | ACE2-Fc [v5] | This paper | | Plasmid to express ACE2 as an human Fc IgG1 fusion protein |
| Recombinant DNA reagent | Spike-mutant | Michael Farzan (Scripps Research, Jupiter, FL) | | Plasmid to express Spike-mutant |
| Recombinant DNA reagent | Spike-delta | This paper | | Plasmid to express Spike-delta on viral coat |
| Recombinant DNA reagent | psPAX2 | Addgene | Cat#: 12260 RRID:Addgene_12260 | |
| Recombinant DNA reagent | pLKO.1 TRC-DsRed | *Buffone et al., 2013* | | |
| Recombinant DNA reagent | VSVG | Addgene | Cat#: 12259 RRID:Addgene_12259 | |

## Materials

Human embryonic kidney 293 T cells ('293T') and Lenti-X 293 T cells were purchased from Clontech/Takara Bio (Mountain View, CA). Stable HEK293T/ACE2 cells were kindly provided by Michael Farzan (Scripps Research, Jupiter, FL). All cells were mycoplasma negative. These cell types were maintained in culture using Dulbecco's Modified Eagle Medium (DMEM) containing 10% fetal bovine serum, 1% Pen-Strep and 1% GlutaMAX supplement. All lectins were from Vector labs. These include VVA (*Vicia Villosa* lectin, binds GalNAcα on O-glycans), SNA (*Sambucus nigra* lectin, binds primarily α2,6 sialic acid), ECL (*Erythrina cristagalli lectin*, binds desialylated lactosamine Galβ1,4GlcNAc chains) and PHA-L (*Phaseolus vulgaris* Leucoagglutinin, binds Galβ1,4GlcNAcβ1,6 (GlcNAcβ1,2Manα1,3)Manα1,3 on complex N-glycans). Goat anti-human ACE2 polyclonal antibody (AF933) and mouse anti-human ACE2 mAb MAB9332 were available from R and D Systems (Minneapolis, MA). Rabbit anti-SARS-CoV-2 RBD (40592-T62) and anti-S2 (40590-T62) pAbs were from Sino Biologicals (Beijing, China). Anti-FLAG clone L5 was from Biolegend (San Diego, CA). All secondary antibodies (Abs) were from Jackson ImmunoResearch (West Grove, PA). When necessary, lectins and Abs were labelled by addition of 25-fold molar excess succinimidyl ester coupled Alexa-405, Alexa-488, Alexa-555 or Alexa-647 dye (Fluoroprobes, Scottsdale, AZ) to protein suspended in phosphate buffered saline (PBS, pH 7.4) for 1 hr at room temperature (RT). Following this, the reaction was quenched with 1/10th volume 1 M Tris, and unreacted Alexa-dye was removed using 7 kDa molecular mass cutoff Zeba desalting spin columns (Thermo). Unless mentioned otherwise, all other reagents were from either Thermo-Fisher or Sigma Chemicals.

## Molecular dynamics simulations

The ACE2-RBD complex structure (6LZG, *Wang et al., 2020*) was superimposed to the trimeric spike-protein state (6VYB *Walls et al., 2020*), in which one of the RBD domains is 'open' so that ACE2 is associated with the RBD domain in the open conformation. Missing residues in the structure were modeled using PyRosetta (*Chaudhury et al., 2010*). N-glycans were then added to the structure using Glycan Reader within CHARMM-GUI (*Jo et al., 2008*; *Jo et al., 2011*). The identity of the glycans was assigned based on a published work (*Watanabe et al., 2020*). The final structure contained glycan modifications at 54 spike Asn and 6 ACE2 Asn.

All simulations were performed using NAMD2 (*Phillips et al., 2005*) using the CHARMM36 parameters (*Vanommeslaeghe et al., 2010*). The complex was first placed in a water box of sufficient size to ensure a minimum 12 Å separation from the wall. Net charge in the molecule was neutralized by adding 841 K+ and 743 Cl- ions, resulting in 867,166 atoms total. Protein and glycan atoms were first fixed while water (TIP3) molecules were energy minimized for 10,000 steps. Next, the entire system was relaxed for another 10,000 steps, and the system temperature was gradually increased to 310 K in increments of 1 K with 120 fs of equilibration at each temperature. A time step of 2 fs was used. Constant temperature and pressure were maintained using the Langevin framework (*Martyna et al., 1994*). Long range electrostatic interactions were treated using the particle mesh Ewald method (*Darden et al., 1993*). The simulation was continued for 26.5 ns. Intermediate structures were saved every 10 ps.

## Molecular biology

A plasmid containing the Spike protein (2019-nCov_pcDNA3.1(+)-P2A-eGFP **[v1]**) was kindly provided by Dr. Haisheng Yu (Institute of Laboratory Animal Science, Peking Union Medical College). To create plasmids **[v3]** and **[v4]**, the full S1 domain coding region (M1 to R682) or RBD coding region (R319 to F541) was PCR amplified and cloned into the NheI/XbaI or AgeI/XbaI sites, respectively of pCSCG-19FcHisP2AdTom (*Lo et al., 2013*). Human ACE2 plasmid was obtained from Addgene (#1786, *Li et al., 2003*). The entire ACE2 coding region was amplified in two segments to silence an EcoRI cutting site in the coding region and joined by overlap extension PCR. The resulting PCR fragment was cloned into the EcoRI/NheI site of pKLV2-CMVpuroBFP vector to obtain [v2]. Additionally, [v5] was created by PCR amplifying the region encoding ACE2 extracellular domain (Q18 to S740) and cloning into the AgeI/XbaI sites of pCSCG-19FcHisP2AdTom plasmid. Plasmids [v2]-[v5] will be distributed via Addgene.

## CRISPR-Cas9 knockout cell lines

HEK293T CRISPR-Cas9 cells were generated by transfecting wild-type 293 T cells with pX330-U6-Chimeric_BB-CBh-hSpCas9 vector (Addgene, Plasmid# 42230) carrying single-guide RNA (sgRNA) targeting either the Core-1 β3Gal-T (*C1GalT1*, target site: GCAGATTCTAGCCAACATAA) or β1,2 GlcNAc-transferase (*MGAT1*, GTGGGGCGCTATCCTCTTTGTGG). While knocking out the former gene results in cells expressing O-glycans truncated at the Tn-antigen stage (GalNAcα±sialic acid), the latter prevents the N-glycan processing beyond the oligomannose/Man5 stage (*Stolfa et al., 2016*). Isogenic clones of both cell lines were obtained by single-cell flow cytometry sorting using fluorescently conjugated VVA and PHA-L for selection. Knockouts were confirmed by Sanger sequencing of genomic DNA. For convenience, cells with disrupted O- and N-glycoprotein processing are termed '[O]⁻293T' and '[N]⁻293T', respectively.

## Recombinant protein expression and purification

All Fc-his tag proteins (vectors [v3], [v4] and [v5]) were expressed by transient transfection of HEK 293 T cells using the calcium phosphate method (*Buffone et al., 2013*), in 3–4 150 mm cell culture Petri dishes per protein. Protein secreted into serum-free media were passed through a 1 mL HisTrap FF column (Cytiva, Marlborough, MA) at 1 mL/min. Following extensive washing with 20 mM sodium phosphate buffer containing 0.5 M NaCl and 10 mM imidazole (pH7.2), the bound protein was eluted by collecting 0.5 mL fractions upon step increasing imidazole concentration in the above buffer to 100 mM (8 mL) followed by 500 mM (8 mL).

Protein concentration in each fraction was determined using a flow cytometry bead assay. To this end, briefly, 5 μm Polybead carboxylate microspheres (Polysciences, Warrington, PA) were

covalently coupled with goat anti-human pAb using 1-Ethyl-3-(3-dimethylaminopropyl)carbodiimide/ EDC chemistry (*Marathe et al., 2008*). 1 µL of each fraction collected during HisTRAP elution was incubated with these beads for 10 min at RT, before washing and addition of 1:200 diluted FITC conjugated secondary Ab for an additional 10 min. Bead-bound Fc-protein was then quantified using flow cytometry. All fractions containing the Fc-fusion proteins were then pooled and spin concentrated to ~150 µL volume. Buffer was exchanged to HEPES buffer (30 mM HEPES, 110 mM NaCl, 10 mM KCl, 10 mM glucose, 2 mM $MgCl_2$) using 7 kDa Zeba desalting columns. Final protein concentration was determined in ELISA format by using a calibrant Fc-protein of known concentration that was verified to be >95% pure based on silver stain analysis. Here, a common anti-human HRP conjugated Ab was used to quantify both the Fc- proteins described in this manuscript and the calibrant Fc in the same assay.

Identity of expressed protein and also viral Spike was determined using western blotting with anti-Fc (Jackson), anti-RBD (Sino Biologicals), anti-S2 (Sino Biologicals) and anti-ACE2 (R and Systems) pAbs. Anti-FLAG mAb L5 was also used for western blotting of Spike-mutant virus as it carried a C-terminal FLAG tag.

## Flow cytometry

Standard flow cytometry methods were applied in some instances to measure cell-surface protein expression and lectin binding, using directly conjugated fluorescent reagents (*Stolfa et al., 2016*). Here the lectins/Abs were added to cells for 15 min, prior to performing a quick wash followed by cytometry analysis. These data are presented as Geometric Mean Fluorescence Intensity ('GMFI').

## Spike-protein-ACE2 binding studies

Full-length Spike-protein and human ACE2 were expressed in 293 T cells (wild-type, [O]⁻293T and [N]⁻293T) by transient transfection using the calcium phosphate method (*Buffone et al., 2013*). These cells are called '293 T/S' or '293T/ACE2' when expressed on wild-type HEK293T cells, and '[O]⁻293 T/S', '[O]⁻293T/ACE2', '[N]⁻293 T/S' or '[N]⁻293T/ACE2' when expressed in the glycosylation-KO mutants. Here, just prior to functional studies, cells were released from the culture substrate 48–72 hr after transfection using PBS containing 5 mM EDTA, washed and re-suspended in HEPES buffer containing 1.5 mM $CaCl_2$. During the binding studies, 0–4 µg/mL S1-Fc, RBD-Fc or ACE2-Fc were added to $20 \times 10^6$ cells/mL at 4°C for 15 min. Following this, 1:200 diluted Alexa-647 (Ax647) conjugated goat anti-human-Fc (Fab')2 Ab was added for an additional 10 min. The cells were then washed, resuspended and read immediately using a 4-laser BD Fortessa flow cytometer. Transfected cells were gated based on EGFP expression for Fc-protein binding studies performed with 293T/S-protein cells, and BFP expression for studies using 293T/ACE2 cells. cell-surface expression of ACE2 and Spike on transiently transfected cells was also monitored on experimental day using flow cytometry. These studies used Alexa-647 conjugated anti-ACE2 (MAB9332, R and D systems) and anti-RBD (Sino Biologicals) antibodies. In some cases, binding studies were also performed with stable 293T/ACE2 cells, in which case fluorescence gating was not necessary.

In some assays, cells were treated with sialidase (200 U/mL *Arthrobacter ureafaciens* α2–3,6,8,9-Neuraminidase, New England BioLabs) for 1 hr at 37°C, and washed using HEPES buffer prior to the binding assay. Sialidase activity was verified based on SNA and ECL staining. In this case, control cells were treated identically, except that sialidase was withheld. SNA was directly conjugated with Alexa-555 and ECL was conjugated with Alexa-647 in order to simplify the implementation of dual color cytometry measurements.

## Pseudovirus production and characterization

SARS-CoV-2 Spike-protein virus was generated by replacing the classical *Vesicular stomatitis* virus (VSV-G) envelope protein of 3rd generation lentivirus with either wild-type Spike protein ('Spike-WT', [v1]) or a mutant Spike plasmid ('Spike-mutant') containing an 'SRAS' sequence in place of furin sensitive 'RRAR' (gift from Michael Farzan, *Quinlan et al., 2020*). In an additional variant called 'Spike-delta', the 'SRAS' site in Spike-mutant was replaced by a single 'A'. To produce these viruses, Lenti-X 293 T cells (Takara Bio, MountainView, CA) were transiently transfected with 17.8 µg of pLKO.1 TRC-DsRed, 22 µg of psPAX2 and 5.8 µg of VSVG (or 9.9 µg of **v1** or 8.6 µg of Spike-mutant/Spike-delta) plasmid in a 15 cm dish using the calcium phosphate method (*Buffone et al., 2013*). Six hours

post-transfection, the medium was changed to virus collection medium (OPTIMEM + 10% FBS). The first batch of virus was collected 18–20 hr thereafter. Virus collection medium was then supplemented with 10 mM sodium butyrate. The second virus batch was collected 18–20 hr later. Both virus batches were pooled, filtered through 0.45 μm filter and ultra-centrifuged at 50,000 × g to concentrate the particles. The viral pellet was resuspended in OPTIMEM + 10% FBS, aliquoted and stored at −80°C until use.

Following the above protocol, 14 additional virus types were generated in this manuscript. This includes virus with envelope composed of: i. VSVG, ii. Spike-WT or iii. Spike-mutant that were produced in either: a. wild-type HEK293T, b. [O]⁻293 T c. [N]⁻293T or d. wild-type HEK293T being cultured in the presence of 15 μM kifunensine (Toronto Research Chemicals, Canada). In addition, Spike-delta was produced in the presence and absence of 15 μM kifunensine. For many of the production lots, viral titer was determined using the lentiviral p24 ELISA kit from Takara Bio (MountainView, CA) following manufacturer's instructions. Virus concentration was also verified by western blotting with anti-S2 pAb or anti-FLAG to detect equivalent amounts of Spike protein in each of the preparations.

## Pseudovirus entry assay

In order to transduce cells, virus was added at various titers indicated in the main manuscript to either wild-type 293Ts, stable 293T/ACE2 cell lines or various glycoengineered variants that transiently overexpressed ACE2 (at 48 hr post-transfection). Such infection was performed in the presence of 8 μg/ml polybrene as described previously (*Buffone et al., 2013*). In some cases, either the virus or cells were treated with sialidase for 1 hr at 37°C under conditions described above, prior to transduction. Virus was removed and fresh medium was added 8 hr post-transduction. DsRed fluorescence was monitored on a daily basis. All flow cytometry and microscopy data presented in this manuscript correspond to the 72 hr time point. Here, DsRed positive cells and the arithmetic mean fluorescence intensity ('MFI') of all cells in the PE-channel were also quantified using flow cytometry (BD Fortessa X-20). Microscopy images were acquired using a Zeiss AxioObserver instrument (10X/0.25 NA or 20X/0.4 NA objective).

## Statistics

All data are presented as mean ± standard deviation. Dual comparisons were performed using the two-tailed Student's t-test. ANOVA followed by the Tukey post-test was used for multiple comparisons. $p < 0.05$ was considered to be statistically significant. Number of repeats are specified in individual panels.

# Acknowledgements

We gratefully acknowledge Dr. Michael Farzan (Scripps Research, Jupiter, FL) for providing HEK293T/ACE2 stable cells and Spike-mutant plasmid, and Dr. Haisheng Yu (Peking Union Medical College, China) for providing Spike-protein plasmid. This work was partially supported US National Institutes of Health grants HL103411, GM133195, GM139160 and GM126537.

# Additional information

### Competing interests

Qi Yang, Thomas A Hughes, Anju Kelkar, Sriram Neelamegham: Co-author of a provisional patent application.(63/079,667). The other authors declare that no competing interests exist.

### Funding

| Funder | Grant reference number | Author |
| --- | --- | --- |
| National Institutes of Health | HL103411 | Sriram Neelamegham |
| National Institutes of Health | GM133195 | Sriram Neelamegham |
| National Institutes of Health | GM126537 | Sriram Neelamegham |

| National Institutes of Health | GM139160 | Sheldon Park |
| | | Sriram Neelamegham |

The funders had no role in study design, data collection and interpretation, or the decision to submit the work for publication.

### Author contributions
Qi Yang, Thomas A Hughes, Sheldon Park, Investigation, Methodology, Writing - review and editing; Anju Kelkar, Formal analysis, Validation, Investigation, Methodology, Writing - review and editing; Xinheng Yu, Kai Cheng, Wei-Chiao Huang, Investigation, Writing - review and editing; Jonathan F Lovell, Conceptualization, Supervision, Writing - review and editing; Sriram Neelamegham, Conceptualization, Formal analysis, Supervision, Funding acquisition, Validation, Methodology, Writing - original draft, Writing - review and editing

### Author ORCIDs
Qi Yang (iD) https://orcid.org/0000-0003-4308-4382
Thomas A Hughes (iD) https://orcid.org/0000-0002-7887-6876
Jonathan F Lovell (iD) http://orcid.org/0000-0002-9052-884X
Sriram Neelamegham (iD) https://orcid.org/0000-0002-1371-8500

### Decision letter and Author response
Decision letter https://doi.org/10.7554/eLife.61552.sa1
Author response https://doi.org/10.7554/eLife.61552.sa2

## Additional files

### Supplementary files
• Transparent reporting form

### Data availability
All data generated or analysed during this study are included in the manuscript and supporting files. All plasmids generated by the authors will be deposited at Addgene.

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
