## [Decision Letter]

**Acceptance summary:**

This work explores the role of N- and O-glycosylation of the SARS-COV-2 Spike protein and its receptor on viral binding and entry into host cells. While changes in the glycosylation of either the Spike protein or the ACE receptor have only modest effects in binding, disruption of N-glycosylation processing was shown to have a profound effect on the internalization and proteolytic processing of the spike protein. Chemical inhibitors of N-glycosylation also inhibited pseudovirus internalization, suggesting that therapeutic targeting of glycoenzymes could be used to reduce viral entry and transmission.

**Decision letter after peer review:**

Thank you for submitting your article "Inhibition of SARS-CoV-2 viral entry upon blocking N-glycan elaboration" for consideration by *eLife*. Your article has been reviewed by three peer reviewers, including Malcolm J McConville as the Reviewing Editor and Reviewer #1, and the evaluation has been overseen by Bavesh Kana as the Senior Editor. The following individual involved in review of your submission has agreed to reveal their identity: Morten Thaysen-Andersen (Reviewer #2).

The reviewers have discussed the reviews with one another and the Reviewing Editor has drafted this decision to help you prepare a revised submission.

This is a very interesting, well-conceived and highly relevant study exploring the role of glycosylation of the SARS-CoV-2 Spike protein and corresponding host ACE2 receptor on viral binding and entry. The authors provide evidence that N- and O-glycosylation of either the viral Spike protein or the host receptor has little effect on viral binding to host cells, but has a dramatic effect on protease cleavage of the Spike protein to allow viral entry.

Summary:

The manuscript explores the impact of N- and O-linked glycosylation on SARS-CoV-2 viral binding to human ACE2 and entry into host cells. Both the viral Spike-protein and ACE2 are known to be heavily glycosylated, yet the functional relevance of the Spike and ACE2 glycans remains largely uncharacterized. In this submission, recombinant expression of several variants of the viral Spike-protein and human ACE2, together with CRISPR-Cas9-driven disruption of Core-1 β3GalT and MGAT1 enzymes were employed to study glycobiology of viral-host interactions. This was combined with the use of chemical inhibitors of key glycan processing enzymes, sialidase treatment, in addition to flow cytometry- and microscopy-based binding/entry assays using pseudoviruses. The study provides robust evidence for the involvement of protein glycosylation in key parts of the viral infection pathway including controlling the furin cleavage rate. On the other hand, the data suggest that Spike and ACE2 protein glycosylation is not required for ACE2-mediated direct binding and viral entry (at least in this model system). The manuscript is highly relevant and timely. The author team is one of the few globally that have the required expertise to study the glycobiology of COVID-19.

1) While the reviewers noted that the authors had undertaken some studies to confirm that the genetic or pharmacological inhibition of N-/O-glycosylation in HEK293 cells had the expected effects on protein glycosylation, the lack of more robust and direct biochemical characterization of the glycosylation changes in the recombinant glycoproteins used in this study (other than the lectin binding studies) was considered a significant shortcoming. For example, strong biochemical validation of the level of sialylation of the ACE2 and Spike proteins expressed in HEK293 cells or effectiveness of the sialidase treatment was lacking. Similarly, validation of the protein-specific changes in glycosylation arising from MGAT1 and C1GalT1 knock-out was also lacking. Further analysis of the S-protein glycoforms on the pseudovirus is warranted for several reasons including:

i) Most analyses of S1 glycosylation to date have been done on recombinant proteins that passes through the secretory pathway with canonical topology. In contrast, the S1 protein embedded in pseudovirus that has budded into the secretory pathway at the ERIC will be in a different membrane, which may alter the extent to which they can be accessed by Golgi glycosyltransferases/glycosidases.

ii) A previous study has reported that levels of O-glycosylation of the S-protein was negligible, contrary to the phenotype observed in this study.

iii) If wild-type Spike and ACE2 carry low levels of sialylation (as expected from the chosen HEK293 cell expression system), then the effect of the sialidase treatment would also be expected to be low. However, sialylation may still play crucial in vivo roles in virus-host interactions in human cells that have higher levels of sialylation than HEK293 cells.

Based on these concerns, it is strongly suggested that the authors provide additional biochemical characterization of the S1 and ACE2 proteins. However, if this is not possible within a reasonable time frame, it is essential that the authors provide a point-by-point discussion of limitations of their current glycan analysis, as well as the use of the HEK293 cells and pseudovirus.

2) In subsection “Knocking out N-glycans on Spike abrogates viral entry” it is mentioned that the Spike pseudovirus with disrupted MGAT1 showed less Spike protein on the surface. This is an important observation that should be expanded on in the manuscript. The altered binding and entry of this variant may arise from either aberrant glycosylation and/or less Spike protein on the surface. The cell surface expression level could be impacted by the glycosylation pattern of the protein. It is important to discuss such points in the manuscript to assist the reader and to leave open the possibility that altered binding and viral entry may be a result of less functional expression of Spike on the protein surface rather than a more direct involvement of the glycans in binding or protection against furin site cleavage.

3) Fc was used as the fusion protein for expression of S1, RBD and ACE2 – this Fc domain would also be glycosylated in these systems – has this been taken into account? Similarly, were the O-links also expressed in this system? Figure 1B does not show whether and where glycosylation was added to the Fc-His expressed proteins.

4) There seems to be some contradiction in the manuscript as to whether the Spike-mutant has increased or decreased proteolysis. Since the furin hydrolysis sequence has been modified to be less susceptible to hydrolysis by the arginine deletions it would be expected to cause decreased hydrolysis of the Spike proteins as is described. However, Figure 4F indicates that the Spike-mutant has increased proteolysis and thus enhanced viral entry. In contrast in Figure 3 the Spike-mutant behaves the same as the wild type in terms of viral entry although it is described as having reduced proteolysis

5) As the data also described O-glycosylation of S1/ACE2 proteins, the title should reflect changes in “protein glycosylation...” rather than just N-glycosylation.

Revisions expected in follow-up work:

6) Biochemical characterization of glycosylation (N-/O-/sialylation) of S1 protein in pseudovirus generated from HEK293 cells following inhibitor treatment or loss of expression of MGAT1 and C1 GatT1.

---

## [Author Response]

Revisions for this paper:1) While the reviewers noted that the authors had undertaken some studies to confirm that the genetic or pharmacological inhibition of N-/O-glycosylation in HEK293 cells had the expected effects on protein glycosylation, the lack of more robust and direct biochemical characterization of the glycosylation changes in the recombinant glycoproteins used in this study (other than the lectin binding studies) was considered a significant shortcoming. For example, strong biochemical validation of the level of sialylation of the ACE2 and Spike proteins expressed in HEK293 cells or effectiveness of the sialidase treatment was lacking. Similarly, validation of the protein-specific changes in glycosylation arising from MGAT1 and C1GalT1 knock-out was also lacking. Further analysis of the S-protein glycoforms on the pseudovirus is warranted for several reasons including:i) Most analyses of S1 glycosylation to date have been done on recombinant proteins that passes through the secretory pathway with canonical topology. In contrast, the S1 protein embedded in pseudovirus that has budded into the secretory pathway at the ERIC will be in a different membrane, which may alter the extent to which they can be accessed by Golgi glycosyltransferases/glycosidases.ii) A previous study has reported that levels of O-glycosylation of the S-protein was negligible, contrary to the phenotype observed in this study.iii) If wild-type Spike and ACE2 carry low levels of sialylation (as expected from the chosen HEK293 cell expression system), then the effect of the sialidase treatment would also be expected to be low. However, sialylation may still play crucial in vivo roles in virus-host interactions in human cells that have higher levels of sialylation than HEK293 cells.Based on these concerns, it is strongly suggested that the authors provide additional biochemical characterization of the S1 and ACE2 proteins. However, if this is not possible within a reasonable time frame, it is essential that the authors provide a point-by-point discussion of limitations of their current glycan analysis, as well as the use of the HEK293 cells and pseudovirus.

We agree with the reviewer comments, and considered these at the time of the original manuscript submission. In particular, we have completed detailed glycoproteomics analysis of S1-Fc, ACE2-Fc and Spike protein from pseudovirus (Cheng et al., manuscript in preparation). The work is still in progress as we are attempting to add the glycoproteomics of the authentic SARS-CoV-2 virus in the same manuscript. We also need to complete some molecular modeling in order to enhance biological relevance. Nevertheless, a summary of these results has been added to the revised manuscript text (Results subsections “Modest role for ACE2 sialic acids during Spike protein molecular recognition” and “Neither sialidase treatment of Spike nor ACE2 markedly impacted viral entry”). As suggested by the reviewers, we have also added a new Figure 3—figure supplement 1 to this manuscript, which contains detailed characterization of the CRISPR-Cas9 KO cell lines using a larger panel of lectins. We now address the 3 specific comments of the editor/reviewers in more detail:

i) Characterization of glycans on Spike (S1-Fc), ACE2 fusion protein (ACE2-Fc) and pseudovirus produced in HEK293T cells

Sialylation pattern of S1-Fc and ACE-2-Fc fusion proteins: We have performed extensive glycoproteomics analysis of these proteins including characterization of site-specific N-glycan mapping (data are freely available from corresponding author upon request, but not provided in this published response letter as it also involves other investigators who are not co-authors of the current manuscript). These runs were performed using an LTQ-Fusion Lumos mass spectrometer (MS) in the laboratory of Prof. Jun Qu (Buffalo, NY, USA) as part of a collaborative project. The experimental workflow involves protein/virus digestion with either the protease trypsin and/or Glu-C under standard conditions (1). Proteins were then subjected to HCD (High-Energy Collision Dissociation or beam-type CID) mode fragmentation. If glycans were observed in this spectrum based on the appearance of signature glycan B-ions, product-dependent CID (Collision induced dissociation) and EThcD (Electron-Transfer/HCD) fragmentation was triggered. All data were processed using GlycoPAT2.0, with FDR (False Discovery Rate) <0.5% and semi-automated inspection to distinguish between isomeric species. Here CID and HCD provide partial information regarding glycan structure and peptide ID. When combined with EThcD a complete “sequencing” of the protein backbone is feasible along with partial characterization of site-specific glycans. Label-free quantitation of each glycopeptide is performed based on measurement of area under the precursor MS^1^ curve (AUC). Using this area and knowledge of glycan structure, it is then possible to classify the relative abundance of different glycan species into 4 groups (see Author response image 1 legends for details). This includes the classification based on: (i) “core glycan type”: core, hybrid, complex or unglycosylated; (ii) “fucosylation type”: core, terminal or non-fucosylated; (iii) “branch type (only for complex glycans)”: mono-, bi-, tri- tetra-antennary; and (iv) “antennae type (only for complex glycans)”: % chains terminated by Gal, GlcNAc or Neu5Ac. (This is a *very* brief description of a complex data analysis framework. Due to the nuances involved, we feel it would be best to publish the glycoproteomics work as a separate manuscript.) Such analysis has been performed for 100s of MS/MS spectra at each of the glycosylation sites, and detailed MS/MS annotation for each is available upon request. This has been performed for ACE2-Fc, S1-Fc, and Spike protein from pseudo(lenti)virus.

In this analysis, we observed that the carbohydrate structures of both S1-Fc and ACE2-Fc contain a mixture of glycans including high-mannose, hybrid and complex structures. Paucimannose structures were not observed, and we are currently investigating the presence of diLacNAc glycans. Among the terminal antennae found on complex glycans, the extent of sialyation varied with protein type, with more sialylation being observed in ACE2-Fc (upto 60% at some sites) compared to S1-Fc (a maximum of 20% at Asn122, Asn147 etc.). Here, *% sialylation* is based on weighting the AUC for each complex glycan by the fraction of antennae what are sialylated.Thus, if a tri-antennary glycan is monosialylated, we only add 1/3^rd^ of the AUC in the % sialylation calculation. It then follows that % sialylated glycans far exceeds % sialylated antennae data presented in the analysis.

Comparison of glycosylation on S1-Fc fusion protein vs. Spike protein of pseudovirus: Similar to the above, we also produced SARS-CoV-2 pseudovirus using HEK293T cells in order to address the exact question raised by the reviewer, regarding potential differences in glycan structures between viral vs. recombinant proteins due to ERGIC trimeric assembly. We then compared a few N-glycosylation sites that were common between the soluble fusion protein (S1-Fc) and the pseudovirus Spike protein. In this analysis, we noted that the type of complex structures (in terms of branching, degree of sialylation etc.) were surprisingly similar in the soluble S1-Fc protein and Spike in virus. However, differences were observed in that Spike protein in pseudovirus tended to have more complex-structures at a given peptide site, compared to S1-Fc which had more high-mannose glycans. This may be due to the nature of viral assembly. Regardless of this subtlety, we observed that the viral coat was extensively sialylated with a high density of complex-type N-glycans. Thus, the absence of sialic acid is not the sole reason for our observation that ACE2-Spike binding proceeds largely in a glycan-independent manner.

With respect to the efficiency of sialidase activity, we are quite confident that these studies were done well. In support of this, a clear 10-fold reduction in SNA binding and 10-fold increase in ECL lectin is reported upon sialidase treatment in Figure 2—figure supplement 1A. All reagents used in this project were fresh and purchased within one month of the experiment. They were carefully aliquoted immediately upon receipt and stored at -20°C.

Overall, the above structural analyses suggests that there is considerable sialylation in both ACE2 and Spike. Perturbing these glycans by desialylation does not impact ACE2-Spike binding mostly because this monosaccharide does not play a major role in binding, except perhaps for a mild steric effect. These conclusions are based on our studies using HEK293T cells, but we would expect them to hold in other systems also. A brief summary of the above results is provided in Results subsections “Modest role for ACE2 sialic acids during Spike protein molecular recognition” and “Neither sialidase treatment of Spike nor ACE2 markedly impacted viral entry”.

ii) Role of O-glycosylation in viral entry

While the glycoproteomics analysis of N-glycans has proceeded well, the assignment of O-glycosylation sites is complicated and is still being finalized. Nevertheless, we have identified a few new O-glycosylation sites on Spike, validated others reported previously (2) and also measured up to 27% occupancy at some locations. Goldman and colleagues also present new data showing that T678 of Spike is glycosylated (3), and this is located adjacent to the R682 furin site. Finally, we have performed small molecule studies with an O-glycan inhibitor and observed inhibition of viral entry. Thus, it appears that O-glycans may impact viral entry though, much needs to be understood as discussed in the revised Discussion. In particular, it remains unknown if the reduced transduction by virus produced in the C1GalT1-KO is a direct effect of changes in O-glycans on Spike, or if this is an indirect effect due to changes in other glycoproteins in the host cell that impact viral assembly and function. These are aspects that we are currently investigating.

iii) Structural characterization of glycans produced in MGAT1 and C1GalT1 cells

As suggested by the reviewers, we have now expanded the characterization of the MGAT1 and C1GalT1-KO cell lines using a larger panel of fluorescent lectins (Figure 3—figure supplement 1). Here we observed that:

1) Many of the lectins that are considered to bind core N-glycan structures (LCA, PHA-L, PHA-E) had low binding to MGAT1-KO cells (both in the presence and absence of sialidase treatment). Their impact on the O-glycan/C1GalT1-KO cells was small.

2) Similarly, many of the lectins that recognize LacNAc structures (PCA, ECL and DSL) displayed low binding to the N-glycan-KO, both in the presence and absence of sialidase treatment.

3) Finally, O-glycan specific lectins (PNA, VVA and SBA) bound differentially to the C1GalT1/O-glycan-KO cells. The binding of these lectins to the MGAT1/N-glycan-KO cells was similar to that of wild-type cells. While we have not performed detailed glycoproteomics of virus produced in these knockout cells, these data and the Sanger sequencing results, are consistent with changes we would expect in *MGAT1* and *C1GalT1* knockout cell lines. The data are also consistent with previous CRISPR-Cas9 mutants from our laboratory (4).

2) In subsection “Knocking out N-glycans on Spike abrogates viral entry” it is mentioned that the Spike pseudovirus with disrupted MGAT1 showed less Spike protein on the surface. This is an important observation that should be expanded on in the manuscript. The altered binding and entry of this variant may arise from either aberrant glycosylation and/or less Spike protein on the surface. The cell surface expression level could be impacted by the glycosylation pattern of the protein. It is important to discuss such points in the manuscript to assist the reader and to leave open the possibility that altered binding and viral entry may be a result of less functional expression of Spike on the protein surface rather than a more direct involvement of the glycans in binding or protection against furin site cleavage.

This comment is not very clear to us, perhaps because one word written as “protein” in the above comment, should have been “viral”? Assuming this, we note that the compounds we are testing are α-mannosidase inhibitors which function after protein folding and after processing via the calnexin/calreticulin pathway. Thus, unlike glucosidase-inhibitors that may impact viral assembly, our compounds do not alter Spike expression on virus surface/coat. We have observed this in our Western blots where protein loading is strictly based on p24 ELISA. We have also performed parallel blots with p24 and anti-Spike to confirm this assertion (data not shown).

Regardless of the above, we agree that our understanding of the impact of glycosylation on SARS-CoV-2 function is incomplete. To continue such examination, we have introduced additional mutations in Spike. In one study, we substituted the “RRAR” furin site with a single “A”, thus resulting in loss of furin based proteolysis (new Figure 5F). The resulting virus displayed robust transduction into 293T/ACE2 cells. Similar furin-independent viral entry has also been reported by others (5). However, surprisingly, we observed that kifunensine could also partially block even this viral entry. Thus, we agree that in addition to furin cleavage regulation, glycans may also have other functions. To investigate this aspect, our laboratory is currently beginning to study the impact of specific glycosylation sites on proteolysis at the Spike S1-S2 interface and also at the S2’ site. Overall, the new data have led us to revise our model (Figure 6C). To reflect this new knowledge, we have also updated portions of Introduction, Discussion and Materials and methods. The basic observation that N-glycans affect viral entry in this system still holds, as we have now repeated this more than a dozen times.

3) Fc was used as the fusion protein for expression of S1, RBD and ACE2 – this Fc domain would also be glycosylated in these systems – has this been taken into account? Similarly, were the O-links also expressed in this system? Figure 1B does not show whether and where glycosylation was added to the Fc-His expressed proteins.

Indeed, one N-glycosylation site was found in the Fc-his section. Similar to the figures by Cheng et al. discussed above, we have classified these into 4 groups in Author response image 1.These panels describe the N-glycan distribution on the Fc glycopeptide of ACE2-Fc (at Asn815, top) and S1-Fc (at Asn757, bottom). Most of the glycans here were bi-antennary, complex type with abundant core-fucose and low levels of sialylation. The glycan structures in both fusion protein Fc sections are similar. Indeed, this mass is accounted for in our data interpretation. The presence of this glycan is also noted in the revised figure legend (see Figure 2A). We also observed one O-glycosylation site on Fc at high (~90%) occupancy. The presence of N- and O-glycans on Fc is incidental, and does not impact the findings presented in this manuscript.

**Author response image 1. respfig1:** Distribution of N-glycans in the Fc-section of ACE2-Fc (top, Asn815) and S1-Fc (bottom, Asn 757). Label-free quantitative analysis helped classify glycans located at individual peptide sites into different sub-groups. This classification was based on: Group 1: relative abundance of high-mannose, hybrid, complex and unoccupied sites on the peptide backbone.; Group 2: relative abundance of fucosylated glycans containing either only core-fucose (i.e. α1,6 linked), terminal-fucose, or both core and terminal fucose. The remaining (glycol)peptide spectra did not contain fucose. Group 3: relative abundance of complex glycans in mono-, bi-, tri and tetra-antennary branched structures; Group 4: Relative abundance of terminal chains on complex glycans, that were capped by either Gal, GlcNAc or Neu5Ac (sialic acid). Each dot is from a single MS run that was either trypsin or Glu-C digested. For each site, we present: i. site of glycosylation, ii. # of PSM analyzed corresponding to this site, iii. Number of unique glycans detected at that location and iv. total number of unique glycopeptides that were analyzed at that site. All identifications are manually verified, and detailed MS/MS annotations are available upon request. Number of PSMs analyzed at each site is provided along with the number of unique glycans at that site. Data are Mean +/- STD for >3 independent LC-MS/MS runs.

4) There seems to be some contradiction in the manuscript as to whether the Spike-mutant has increased or decreased proteolysis. Since the furin hydrolysis sequence has been modified to be less susceptible to hydrolysis by the arginine deletions it would be expected to cause decreased hydrolysis of the Spike proteins as is described. However, Figure 4F indicates that the Spike-mutant has increased proteolysis and thus enhanced viral entry. In contrast in Figure 3 the Spike-mutant behaves the same as the wild type in terms of viral entry although it is described as having reduced proteolysis

We apologize for this confusion. Indeed, Spike-mutant virus/protein has decreased proteolysis and increased (~6-10-fold higher) viral entry into 293T/ACE2 cells (see Figure 2F). There is no contradiction in the data, only miscommunication. We believe the problem may have risen because the previous Figure 4F was too complicated. Thus, we have taken effort to simplify this conceptual model and the new model is presented in Figure 6C. We have also rewritten a portion of the Discussion section to clarify this point. This section describes our data and also data from other laboratories that support the “inverse relation between furin cleavage and viral entry” in 293T/ACE2 cells and also other cell systems.

5) As the data also described O-glycosylation of S1/ACE2 proteins, the title should reflect changes in “protein glycosylation…” rather than just N-glycosylation.

We agree that O-glycans may also play a role in SARS-CoV-2 viral entry, although the exact mechanisms needs to be established. This is clarified in the revised title.

Revisions expected in follow-up work:6) Biochemical characterization of glycosylation (N-/O-/sialylation) of S1 protein in pseudovirus generated from HEK293 cells following inhibitor treatment or loss of expression of MGAT1 and C1 GatT1.

These data are presented and discussed above (please see #1 above).

**References**

1. Liu G, Cheng K, Lo CY, Li J, Qu J, Neelamegham S. A Comprehensive, Open-source Platform for Mass Spectrometry-based Glycoproteomics Data Analysis. Mol Cell Proteomics. 2017;16(11):2032-47.

2. Shajahan A, Supekar NT, Gleinich AS, Azadi P. Deducing the N- and O- glycosylation profile of the spike protein of novel coronavirus SARS-CoV-2. Glycobiology. 2020.

3. Sanda M, Morrison L, Goldman R. N and O glycosylation of the SARS-CoV-2 spike protein. bioRxiv. 2020.

4. Stolfa G, Mondal N, Zhu Y, Yu X, Buffone A, Jr., Neelamegham S. Using CRISPR-Cas9 to quantify the contributions of O-glycans, N-glycans and Glycosphingolipids to human leukocyte-endothelium adhesion. Sci Rep. 2016;6:30392.

5. Hoffmann M, Kleine-Weber H, Pohlmann S. A Multibasic Cleavage Site in the Spike Protein of SARS-CoV-2 Is Essential for Infection of Human Lung Cells. Mol Cell. 2020;78(4):779-84 e5.